# Evaluating the health effect of a Social Housing programme, Minha Casa Minha Vida, using the 100 million Brazilian Cohort: a natural experiment study protocol

Andrêa J F Ferreira ![ORCID],[1,2] Julia Pescarini ![ORCID],[3] Mauro Sanchez,[4] Renzo Joel Flores-Ortiz,[5] Camila Silveira Teixeira,[1] Rosemeire Fiaccone,[6] Maria Yury Ichihara,[1] Rodrigo Oliveira,[7] Estela M L Aquino,[1] Liam Smeeth,[8] Peter Craig,[9] Sanni Ali,[10] Alastair H Leyland,[9] Mauricio L Barreto,[1] Rita de Cássia Ribeiro,[1] Srinivasa Vittal Katikireddi ![ORCID] [11]

RdCR and SVK are joint senior authors.

**Correspondence to**
Andrêa J F Ferreira;
andreaferreiracv@gmail.com

## ABSTRACT

**Introduction** Social housing programmes have been shown to influence health, but their effects on cardiovascular mortality and incidence of infectious diseases, such as leprosy and tuberculosis, are unknown. We will use individual administrative data to evaluate the effect of the Brazilian housing programme Minha Casa Minha Vida (MCMV) on cardiovascular disease (CVD) mortality and incidence of leprosy and tuberculosis.

**Methods and analysis** We will link the baseline of the 100 Million Brazilian Cohort (2001–2015), which includes information on socioeconomic and demographic variables, to the MCMV (2009–2015), CVD mortality (2007–2015), leprosy (2007–2015) and tuberculosis (2007–2015) registries. We will define our exposed population as individuals who signed the contract to receive a house from MCMV, and our non-exposed group will be comparable individuals within the cohort who have not signed a contract for a house at that time. We will estimate the effect of MCMV on health outcomes using different propensity score approaches to control for observed confounders. Follow-up time of individuals will begin at the date of exposure ascertainment and will end at the time a specific outcome occurs, date of death or end of follow-up (31 December 2015). In addition, we will conduct stratified analyses by the follow-up time, age group, race/ethnicity, gender and socioeconomic position.

**Ethics and dissemination** The study was approved by the ethic committees from Instituto Gonçalo Muniz-Oswaldo Cruz Foundation and University of Glasgow Medical, Veterinary and Life Sciences College. Data analysis will be carried out using an anonymised dataset, accessed by researchers in a secure computational environment according to the Centre for Integration of Data and Health Knowledge procedures. Study findings will be published in high quality peer-reviewed research journals and will also be disseminated to policy makers through stakeholder events and policy briefs.

## Strengths and limitations of this study

► This will be the first study to evaluate the effect of a major social housing programme on health outcomes in a middle-income country and is likely to be the largest of its type across the world. This would allows to assess impacts on population subgroups, including adoption of an intersectionality perspective.

► A comprehensive assessment of health impacts is being conducted, including both infectious and noncommunicable diseases (NCDs).

► Our analytical approach includes the use of Propensity Score Matching, which has the limitation of accounting for only observed variables.

► Health and behavioural information (such as smoking status, diet and comorbidities) prior to the intervention are not available, and there is therefore a risk of residual confounding arising from inadequate comparability between exposure groups.

► Finally, this study does not allow estimating long-term effects of Minha Casa Minha Vida on health, especially for NCDs, such as cardiovascular disease mortality and neglected diseases, such as leprosy, given the limited length of follow-up available (up to 8 years).

## INTRODUCTION

Housing is a basic human right and an important social determinant of health and well-being, which not only includes the guarantee of shelter, but also its quality.[1 2] Several studies have investigated the relationship between population health and housing conditions, most of them in high-income countries.[1] [3–6] Residential instability, crowding, temperature and substandard housing conditions (such as water leaks,

poor ventilation, dirty carpets and pest infestation) are associated with chronic and infectious diseases.[1 2 7 8] The neighbourhood in which a house is located has also been shown to be associated with health in studies in high-income countries.[9] Physical neighbourhood characteristics that have been associated with health outcomes include: green parks, schools, health services, sidewalks, public transport, sanitation, aesthetic characteristic and connectivity of the street, bike lanes, availability and relative cost of healthy foods and tobacco.[5 9–11] Less visible but also important are social neighbourhood characteristics, which include measures of social network and support, violence and social capital, especially in vulnerable communities in high-income countries.[3 4 12]

Taking into account these relationships, there is a policy expectation that housing interventions could contribute to improve health and reduce social inequalities, especially among the most vulnerable.[1 5 13] Despite this, we are aware of little or no robust evidence on the positive and negative effects of housing conditions on cardiovascular disease (CVD) and infectious diseases, such as leprosy and tuberculosis. Understanding housing impacts on health in low-income and middle-income countries also remains particularly poorly understood.

In Brazil, there is a double burden of infectious and chronic diseases among the poorest people and, therefore, we will evaluate the effect of Minha Casa Minha Vida (MCMV), a social housing programme, both on mortality from CVD, the leading cause of death in the country, and on infectious diseases associated with poverty, like tuberculosis and leprosy.[14 15] In Brazil, 27% of all-cause of death is attributed to CVD, with most assigned to ischaemic heart disease and stroke.[16 17] Leprosy and tuberculosis are two of the most important infectious diseases in Brazil and affect predominantly vulnerable and marginalised populations.[18–20] Brazil has the second highest leprosy incidence worldwide, with almost 30 000 cases annually.[14] Tuberculosis is also common; the country reported 72 788 new cases in 2018 and 4534 deaths in 2017.[14]

Given the social patterning of housing access and quality in Brazilian context, housing interventions may also have important impacts on the health of disadvantaged population groups, with potential benefits for racial/ethnic minorities, women and individuals and families of lower socioeconomic position. We, therefore, aim to evaluate the health effects of the Brazilian social housing programme MCMV, the largest social housing programme in Latin American.

Our detailed objectives are:

1. To estimate the effect of MCMV on premature CVD, ischaemic heart disease and cerebrovascular disease mortality.
2. To estimate the effect of MCMV on all-cause mortality.
3. To estimate the effect of MCMV on leprosy and tuberculosis incidence.
4. To investigate whether any observed effects of MCMV on cardiovascular and all-cause mortality, leprosy and tuberculosis incidence differ by population subgroups (gender, race/ethnicity, age, socioeconomic position and length of follow-up).

## METHODS AND ANALYSIS
### Study design and population
This is a dynamic, retrospective and open cohort study that will be drawn from individuals registered in the baseline of The 100 million Brazilian Cohort,[21 22] a cohort of individuals applying for government social programmes in Brazil.

### Patient and public involvement
This research was done without public involvement. Public were not invited to comment on the study design and were not consulted to develop public relevant outcomes or interpret the results, since we use an administrative and deidentified dataset and do not have permission to contact individuals. Study findings will be discussed with managers and specialists from the National Housing Secretariat from the Ministry of Regional Development and the published results will be disseminated to the public through the mass media. This study is a joint effort with the National Housing Secretariat from the Ministry of Regional Development, in order to guarantee that the findings would answer relevant policy questions. Centre for Integration of Data and Health Knowledge staff are, in synergy with these key stakeholders and decision-makers, providing the methodological rigour needed to assure sound results. Findings will be incorporated into the National Housing Plan which is currently under development in Brazil. The National Housing Secretariat from the Ministry of Regional Development will not interfere in the analysis and results from studies planned in this protocol.

### Intervention
We report intervention characteristics as suggested by the Template for Intervention Description and Replication for Population Health and Policy (TIDieR-PHP) template.[23] MCMV was implemented in July 2009 by the Brazilian Federal Government.[24] Its main goals are to reduce the housing shortage in Brazil (which exceeded 6 million houses in 2016), with 89% of unmet need concentrated among low-income families (defined as earning less than three times the minimum wage), and improve the construction sector through job generation and wider Brazilian economic growth.[24] By 2018, the government had contracted 5 164 075 and delivered 3 787 200 million house units, resulting in over six million people receiving housing across Brazil.[25]

MCMV was structured to reach families from different income classes using three distinct eligibility criteria and subsidies. In this study, we focus on class 1 subsidies, which targets low-income families, defined as households with less than three times the minimum wage (R$6220 in 2010, equivalent to US$11 6 25) per month. Class 1 subsidies are divided in two subprogrammes targeting either urban or rural areas. In municipalities with more

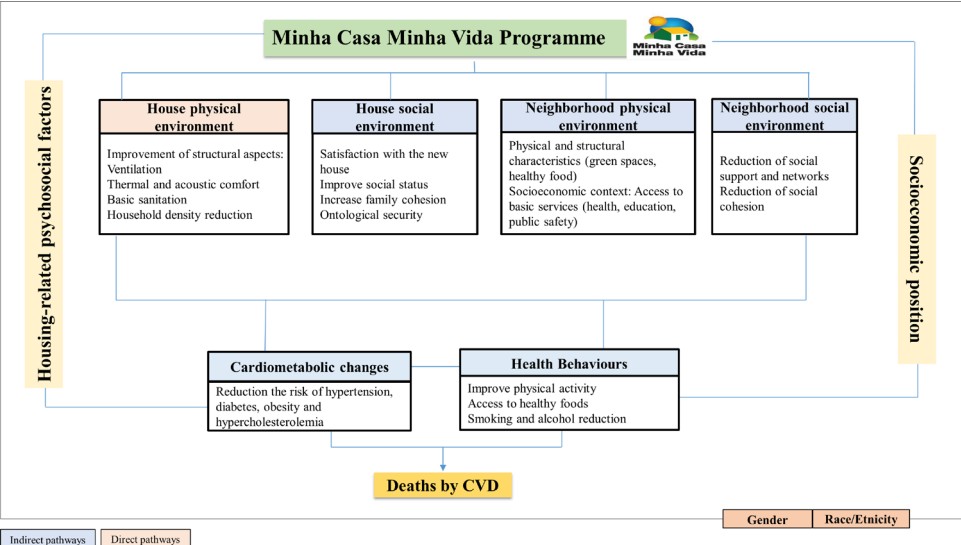

**Figure 1** Logic model to evaluate the effect of Minha Casa Minha Vida on reduction of cardiovascular mortality. CVD, cardiovascular disease.

than 50 000 inhabitants, the MCMV programme uses the Residential Lease Fund (FAR-MCMV) to build or acquire new housing units. Individuals eligible for class 1 MCMV living in municipalities with less than 50 000 inhabitants or living in rural areas receive other forms of MCMV adapted to their context. For this study, we focus on FAR-MCMV, which is the largest subprogramme of MCMV. From 2009 to 2015, we estimate that FAR-MCMV delivered over 1.2 million house units across the country.[25]

Eligibility criteria for FAR-MCMV include: (1) the applicant must have a household income less than or equal to three times the minimum wage (without considering other social benefits, such as income from a conditional cash transfer programme); (2) not be an owner, assignee or promising buyer of a residential property and (3) not have received any previous housing benefits or grants for the purchase of construction materials.[24] Priority criteria for FAR-MCMV include: (1) living in a hazardous or unhealthy area or being homeless; (2) belonging to a family headed by a lone mother (ie, no male partner); (3) having a disabled person(s) in the household, with legal proof and (4) having elderly people, aged 60 years old or over within the household.[24]

### Logic models

We created a logic model for each of the health outcomes studied in this protocol, informed by the existing literature, to describe the hypothesised mechanisms through which the MCMV may affect (1) CVD mortality (figure 1), (2) leprosy new case detection (here and after, named as 'incidence') (figure 2) and (3) tuberculosis incidence (figure 3). We identified, for each of the outcomes, pathways that are likely to operate through direct (physical house conditions) and indirect forms (housing neighbourhood effects and subjective aspects associated with house ownership).

### Logic model for CVD mortality

The Programme may affect cardiovascular health through different pathways. Possible direct effects include changes in the physical standards of housing, leading to improvements in thermal and acoustic comfort, basic sanitation and household density reduction.[2 26–28] Indirect effects are related to the inclusion of families in new social and physical neighbourhood environments—due to the relocation process—with potentially better socioeconomic, structural and physical contexts, as well as better access to basic health services.[10 29 30] On the other hand, relocation may negatively influence the social environment of the neighbourhood, since beneficiaries lose social networks and support, reducing social cohesion[3 12 31] (figure 1).

In the short term, improvements in living conditions could enable changes in health behaviours, like physical activity, diet, alcohol and tobacco consumption.[4 11] In addition, reducing the cost of housing also provides greater access to resources which can be spent on healthier food and healthcare, leading to better control of cardiometabolic risk factors.[32 33] In contrast, it is possible that greater availability of income could lead to greater consumption of unhealthy products (such as ultraprocessed food, tobacco and alcohol) which could in turn increase cardiovascular risk[34 35] (figure 1).

In the long term, the programme could contribute to reducing chronic and cumulative exposure to psychosocial risk factors arising from inadequate housing contexts and this could potentially reduce the incidence of cardiovascular events.[36 37] The effects of the programme on cardiovascular outcomes might be differential by age, race/ethnicity, socioeconomic position and gender, and might contribute to the reduction of disparities in cardiovascular mortality in Brazil (figure 1).

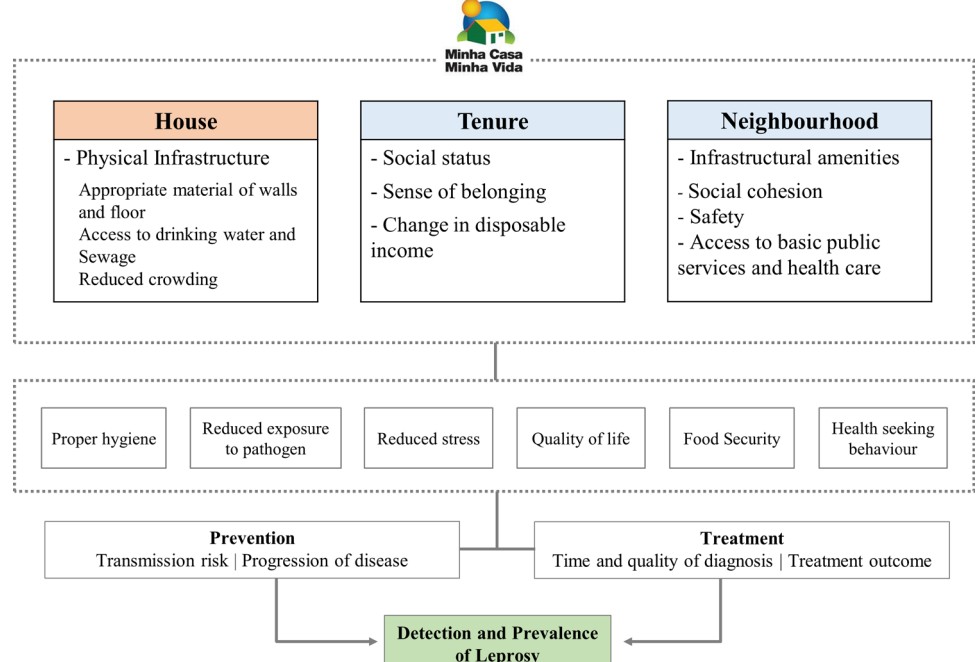

**Figure 2** Logic model to evaluate the effect of Minha Casa Minha Vida on Leprosy Incidence.

## Logic model for leprosy

There is a strong relationship between housing and differential leprosy exposure.[38–41] Housing with better infrastructure and access to drinking water and adequate sanitation may improve hygiene conditions.[18 42] Reduction in household crowding leads to lower contacts among members of the family and, consequently, reduction of leprosy transmission[43 44] (figure 2).

Access to neighbourhood amenities might improve access to health services, as well as improve community cohesion, which are important influences on leprosy risk.[38 45] Places to purchase healthy food can facilitate food security, which might result in improvements in the nutritional status of individuals and accelerate the immune response in leprosy infection[38 46] (figure 2).

Housing ownership may give feelings of security and/or prestige. Also, it can provide greater availability of income for expenditure on potentially health-enhancing products such as food and healthcare.[13 47–49] It is known that access to healthcare is important to support strategies for self-care, case detection, timely diagnosis and treatment, and prevention of more severe forms of the disease[50 51] (figure 2).

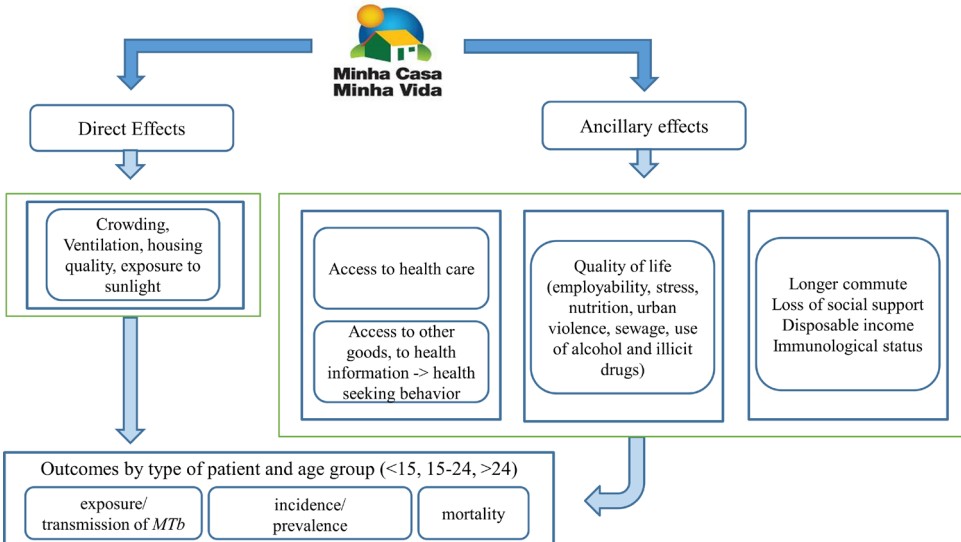

**Figure 3** Logic model to evaluate the effect of Minha Casa Minha Vida on tuberculosis incidence. MTb, *Mycobacterium tuberculosis*.

## Logic model for tuberculosis

The programme may affect tuberculosis incidence through different pathways. Better housing leads to better ventilation and overall housing conditions, such as exposure to sunlight and reduced crowding, which reduce the possibility of transmission through aerial dispersion of the bacillus.[52 53] Alternatively, we may observe beneficiaries experiencing better quality of life or socioeconomic status due to a better environment.[53] Better housing location, if accompanied by more access to public services, including healthcare, can alter individual susceptibility to disease and improve treatment outcomes for those already infected. On the other hand, longer distances to school and workplace may lead to physical stress, worsened immunity and predispose people to infection given exposure to *Mycobacterium tuberculosis*.[20 54] Social support may also change due to the relocation, and may affect exposure to the agent as well as disease progression[55] (figure 3).

We expect a differential effect of this housing programme by age group as the direct effects of the housing material will be more pronounced in children, especially in the case of intradomiciliary contacts of tuberculosis cases. As people grow older and become more exposed to external environment related factors, we hypothesise the effect of the programme to be the net effect of improved housing quality and surrounding area related factors (figure 3).

## Datasets

We will link deterministically the baseline of the 100 Million Brazilian Cohort (2001–2015),[56] which includes information on socioeconomic and demographic variables, to the FAR-MCMV (2009–2015), the mortality information system (SIM) (2007–2015), and the leprosy (2007–2015) and tuberculosis (2007–2015) registries from the Notifiable Disease Information System.[22 57 58] The final deidentified dataset will contain information from recipients and non-recipients of the FAR-MCMV.

## Sociodemographic variables

The 100 million Brazilian Cohort includes baseline socioeconomic and demographic data from over 114 million individuals (approximately 55% of the Brazilian population in 2019). The cohort comprises people who are enrolled in the *Cadastro Unico*, a register containing all individuals within a household that have applied for any social programme administered by the Brazilian Federal Government.[22] We extracted individual level information (age, sex, race/ethnicity, education and occupation), household characteristics (household density, region and area of residence, household construction material, water supply, sanitation, electricity and waste disposal); and monthly per capita income for all family members.[22]

## MCMV programme: FAR modality

Socioeconomic and demographic information (date of birth, gender, marital status, household monthly per capita income, name of head of household) from those who sign the contract to receive the housing unit—the main beneficiary—is obtained from the FAR-MCMV database.[22] In addition, we will extract from this database the name and Social identification number from those who sign the contract, the date of signature of the contract (ie, a proxy for the time that housing unit was delivered to the family), individuals who live with the main beneficiary at that time, and the address of the housing unit or the condominium delivered.[22]

## Mortality information system

Deaths within Brazil are subject to certification by medical professionals, so the causes of death (using International Classification of Diseases, version 10 (ICD-10) codes) can be ascertained reasonably precisely. Despite the death registry being compulsory, there is evidence of underascertainment of deaths, particularly within areas that are more rural and in poorest regions of Brazil.[59] Coding of CVD-related causes is thought to be good, but for some analyses, corrections are needed to take account of ill-defined causes.[60] In addition, since 2010 the proportion of garbage codes, causes of deaths that should not be considered as underlying causes of deaths, had declined, and in 2015, an estimation, show that 97.2% of deaths were included in the mortality system.[60] The highest ascertainment was observed in the state of São Paulo (99.8%), located in the Southeast region, and the lowest was observed in the state of Amapá (91.2%), in the North region of the country.[60] Underascertainment and miscoding of deaths is known to be most problematic in older adults, as well in young children.[60]

## Notifiable diseases information system

Leprosy and tuberculosis notification to the notifiable diseases information system (SINAN) disease registry is compulsory in Brazil and coded using ICD-10. The records include individuals' sociodemographic and clinical information at the time of diagnosis and treatment updates when available. SINAN has improved its quality and completeness over time.[61] However, under-reported cases and missing information still happen, especially in the poorest regions of the country. Therefore, leprosy and tuberculosis reporting to the SINAN notification system is based on passive surveillance and there is therefore heterogeneity in the frequency and completeness of reporting, which may result in the true incidence of diseases being underestimated.[61]

## Data analysis plan
### Definitions of exposure and outcomes

We will define our exposed population as individuals who signed the contract to receive the housing unit from FAR-MCMV and the household members who live with them at the time of the contract signature. If there are no household members registered at the time of contract signature, we will include household members that appear in

the last update of *Cadastro Unico* carried out up to 2 years before receipt of the house unit from FAR-MCMV.

All outcomes will be defined according to the ICD-10. We will evaluate the following outcomes: (1) all-cause mortality; (2) CVD mortality (I00-I99); (3) ischaemic heart disease mortality (I20–I25); (4) cerebrovascular disease mortality (I60–I69); (5) incidence of leprosy (A30) and (6) incidence of tuberculosis (A15–A19). All-cause and CVD mortality will be evaluated in adults aged 30–69 years old (defined as premature mortality within Brazil and of particular relevance to health policy) and in adults aged 18–69 years old (the broader adult population with the most reliable mortality data). We will not investigate outcomes among older adults, given the known issues of ascertaining mortality for this age group within the SIM. The incidence of leprosy and tuberculosis will be evaluated in all age groups.

To evaluate all-cause and specific cardiovascular mortality in individuals aged 18–69 years old, the follow-up time (in years) for each individual will start at entry into the analytical cohort (ie, on signing a contract for the intervention group or the matching date for the control group) or the age at which an individual reaches 18 years of age, whichever is later. The follow-up time will end at the first of: date of death, end of follow-up (31 December 2015) or reaching 70 years of age.

To evaluate the incidence of leprosy and tuberculosis among individuals for all ages, the follow-up time (in years) for each individual will start at entry into the analytical cohort and will end at the first diagnosis of leprosy or tuberculosis, date of death or end of follow-up (31 December 2015).

### Analysis

Estimating the effect of the FAR-MCMV programme on health outcomes is challenging due to selective uptake of the intervention by individuals. To address this, we will use different Propensity Score (PS) approaches to identify comparable individuals (based on observable characteristics) who did and did not receive the FAR-MCMV intervention, given individual-level characteristics.[62] Matching methods will include nearest neighbour matching using narrow callipers to minimise bias and Kernel matching.[63 64] In addition to matching methods, we will also estimate the effect of MCMV on the selected outcome using survival models weighted by the Inverse Probability of Treatment to estimate the average treatment effects and average treatment effect on the treated.[62]

In addition, we will stratify the analyses for key subgroups of interest—namely, gender, race/ethnicity, socioeconomic position and phases of implementation of MCMV programme. We will also investigate whether effects differ across combinations of these characteristics. Specifically for leprosy, we will also stratify the analyses by residence in a high leprosy burden municipality as defined by the Brazilian Ministry of Health.[65] We will also stratify our analyses for individuals that are (and are not)

beneficiaries of the Bolsa Familia Programme, one of the largest conditional cash transfer of the world.[66]

### Robustness checks

To check the robustness of our findings we will perform different tests. First, in the PS matching analysis, we will restrict our data from matching individuals with more narrow PS (different callipers). We will also restrict our analysis to certain types of municipality where data from MCMV-FAR or the mortality information system have better quality.

### Sensitivity analyses

The matching strategy relies on the outcome being independent of treatment, conditional on the PS (Conditional Independence Assumption (CIA)).[67] However, if there are unobserved variables which affect assignment into treatment and the outcome variable simultaneously, a bias might arise. Since this assumption is non-testable by its nature, questions about the plausibility of the CIA can arise, and our results or at least their statistical significance could probably be driven by an omitted variable strongly correlated with the treatment outcome.

We will carry out sensitivity analyses to assess how strong the influence of these postulated unobservables might be in our study. We will use Rosenbaum bounds approach[68] and the sensitivity strategy proposed by Ichino *et al*.[67] This approach aims to assess the bias of our estimates when the CIA is assumed to fail in some meaningful way. A failure in the CIA is equivalent to saying that the assignment to treatment is not unconfounded given the set of observable variables.[67]

## ETHICS AND DISSEMINATION

The 100 million Brazilian Cohort study was approved by the ethics committee of Instituto Gonçalo Muniz-Oswaldo Cruz Foundation (project number: 1.612.302) and the specific aims of this project was submitted for ethical approval in the same ethics committee. In addition, the University of Glasgow Medical, Veterinary and Life Sciences College Ethics Committee also approved the study (project number: 200190001). All data are linked in a safe room with access restricted to specified people only. After the data are linked and the linkage accuracy is calculated, researchers will have full access to the deidentified dataset. The dataset will be accessed by researchers on application to a data curation committee with a detailed analysis plan. The dataset will receive a digital object identifier, and full specification of how the dataset was created will be available online. All manuscripts will be published in high quality peer-reviewed open access journals and will also be disseminated to policy-makers through stakeholder events and policy briefs.

**Author affiliations**
[1]Public Health Institute, Federal University of Bahia, Salvador, Brazil
[2]Centro de Integração de Dados e Conhecimentos Para Saúde (Cidacs), Fiocruz Bahia, Salvador, Brazil

[3]Centro de Integração de Dados e Conhecimentos Para Saúde (Cidacs), Fundação Oswaldo Cruz, Salvador, Brazil
[4]Public Health, Universidade de Brasília, Brasilia, Brazil
[5]Center for Integration of Data and Health Knowledge (Cidacs), Fiocruz Bahia, Salvador, Brazil
[6]Mathematics and Statistics, Universidade Federal da Bahia, Salvador, Brazil
[7]Economy, Federal University of Bahia, Salvador, Brazil
[8]Epidemiology and Population Health, London School of Hygiene and Tropical Medicine Faculty of Public Health and Policy, London, UK
[9]Public Health Sciences Unit, University of Glasgow MRC/CSO Social and Public Health Sciences Unit, Glasgow, UK
[10]Department of Non-communicable Disease Epidemiology, London School of Hygiene & Tropical Medicine, London, UK
[11]MRC/CSO Social & Public Health Sciences Unit, University of Glasgow School of Life Sciences, Glasgow, UK

**Contributors** SVK, AJFF and JP wrote the first draft of the protocol. RdCR, RJF-O, CST, MS, RF, MYI, RO, EMLA, SA, PC, LS, AHL and MLB contributed with additional material. All authors reviewed the final version.

**Funding** This research was funded by the National Institute for Health Research (NIHR) (GHRG /16/137/99) using UK aid from the UK Government to support global health research. The Social and Public Health Sciences Unit is core funded by the Medical Research Council (MC_UU_12017/13) and the Scottish Government Chief Scientist Office (SPHSU13). SVK is funded by a NHS Research Scotland Senior Clinical Fellowship (SCAF/15/02). Cidacs/Fiocruz is supported by grants from CNPq/MS/Gates Foundation (401739/2015-5), the Wellcome Trust, UK (202912/Z/16/Z).

**Disclaimer** The views expressed in this publication are those of the authors and not necessarily those of the NIHR, the UK Department of Health, CNPq/MS/Gates Foundation and the Wellcome Trust.

**Competing interests** None declared.

**Patient and public involvement** Patients and/or the public were not involved in the design, or conduct, or reporting, or dissemination plans of this research.

**Patient consent for publication** Not required.

**Provenance and peer review** Not commissioned; externally peer reviewed.

**ORCID iDs**
Andrêa J F Ferreira http://orcid.org/0000-0002-6884-3624
Julia Pescarini http://orcid.org/0000-0001-8711-9589
Srinivasa Vittal Katikireddi http://orcid.org/0000-0001-6593-9092

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
