## [Reviewer comments · BMJ Open]

ARTICLE DETAILS

TITLE (PROVISIONAL)	Evaluating the health effect of a Social Housing programme, Minha Casa Minha Vida, using the 100 million Brazilian Cohort: A natural experiment study protocol
AUTHORS	Fortes Ferreira, Andr�ea Jacqueline; Pescarini, Julia; Sanchez, Mauro; Flores-Ortiz, Renzo; Teixeira, Camila; Fiaccone, Rosemeire; Ichiara, Maria Yury; Oliveira, Rodrigo; Aquino, Estela M. L.; Smeeth, Liam; Craig, Peter; Ali, Sanni; Leyland, Alastair; Barreto, Mauricio; Ribeiro, Rita de C�assia; Katikireddi, Srinivasa

VERSION 1 – REVIEW

REVIEWER	Dr Tim Taylor University of Exeter Medical School, Truro, UK Previously principal investigator and currently Co-I on SMARTLINE project funded by ERDF. SMARTLINE is examining health and wellbeing determinants and looking at interventions in a cohort in social housing in Cornwall UK.
REVIEW RETURNED	17-Aug-2020

GENERAL COMMENTS	This is an interesting paper outlining a protocol for a study on the impact of a social housing intervention in Brazil. The study uses secondary data analysis of the novel merger of datasets from a large cohort study, a major social housing intervention and health datasets available in the country. The data analysis protocol appears robust, with appropriate plans for the identification of comparable individuals using appropriate data on location and socioeconomic status. Major comments 1. Page 3 – lines 10-15. I am always a bit nervous about “firsts”. Here you are claiming to be the “first study to evaluate the effect of a social housing programme on health outcomes at national level”. I don’t dispute that what you are planning is important and that it is novel, but I wonder about this statement. There is work using the SAIL database in Wales on certain interventions that have been made on housing – some of which largely fall in the area of improving social housing – e.g. the Housing Regeneration and Health Study. Admittedly this study focuses on improving existing social housing as the intervention, rather than the creation of new housing. There are also studies in Australia (e.g. Bentley et al, 2018) on mental health and social housing. A systematic review of the literature from 1887 to 2007 shows some studies that cover the provision of new housing (Thomson et al, 2009). While what you are doing is a major advance, I wonder if you might focus on scale and the low- to mid-income focus in terms of the novelty of the study on lines 10-16.
---

	2. You suggest there is a lack of patient and public involvement in the research (Page 5, lines 42-47). I am surprised you are not including any public involvement in the research process – for dissemination if not for research design. Do your stakeholders mentioned in line 39 of the Ethics summary not include the public or representative groups? If there is truly no public engagement then it might be worth putting something in lines 42-47 on why this is (perhaps funding constraints?) and it might be worth thinking of attempting to obtain further funding to support such engagement. It is all too easy to analyse data on households without engaging with the people in them, which can give a richer understanding of the reasons why the relationships are as they come out or of what questions could be posed in the research. 3. P7 – I am wondering if you might consider increasing the consideration of potential short term negative impacts of the programme – e.g. if relocation leads to increased stress, fragmentation of social networks (which you do mention briefly) and the wider consequences for households of such a move. I think you may also be being optimistic that reduced housing costs would lead to disposable income being spent on healthy food. Minor comments  1. Page 3 Line 16 – meaning of “intersectionality” not clear to me. “Intersectoral” or “interdisciplinary” might be better words. 2. I think you are overstating weakness on longer term outcomes for bullet 5 (Page 3, line 31) – to have up to 8 years is not exactly limited (often studies are shorter in their focus than this in terms of follow-up). I would consider rephrasing this as a strength. 3. Page 4 Line 37 – add “and infectious disease, such as leprosy and tuberculosis” 4. Page 5, line 24 insert “and” at end of line References Bentley, R. et al (2018) “The impact of social housing on mental health: longitudinal analyses using marginal structural models and machine learning-generated weights”. International Journal of Epidemiology Volume 47, Issue 5, pp 1414-1422. Thomson, H. et al (2009) “The Health Impacts of Housing Improvement: A Systematic Review of Intervention Studies From 1887 to 2007” American Journal of Public Health, Supplement 3, Vol 99, No S3, pp S681-S692.
--	---

REVIEWER	Rebecca Bentley University of Melbourne, Australia
REVIEW RETURNED	08-Sep-2020

GENERAL COMMENTS	This study protocol for evaluating the effect of a social housing program in Brazil using a natural experiment methodology will enable quantification of the health protective potential of social housing. This is an important and worthwhile area of investigation that is difficult to research because of the composition of residents in social housing often being different to other tenures. It proposes to use cohort data, that includes information on tenancy, deterministically linked to registry data to examine time spent in social housing in relation to cardiovascular mortality, leprosy and tuberculosis (chosen to cover both chronic and infection disease pathways). The cohort in social housing will be matched to similar
--

	individuals in other housing tenures using propensity score approaches. The proposed project will make efficient use of existing cohort and registry data. Importantly, because of the large sample size, 114 million people or 55% of the Brazilian population, the experiences of subgroups defined to capture the complexity of identity (intersectionality) can be explored. There is a lack of information on important behavioural risk factors (e.g. smoking), however, these factors (while correlated with social housing) are unlikely to determine access to the program. Authors acknowledge that the maximum follow-up is 8 years, which is a short time window to examine the chronic pathway under consideration. Pathways are explicitly described in logic models, however, these models (unlike directed acyclic graphs) do not identify sources of confounding, the direction of relationships, mediators and modifiers which would be useful for framing the approach more clearly (particularly if complex sources of heterogeneity are to be explored). Sensible robustness checks and sensitivity analyses are proposed.
--	--

VERSION 1 – AUTHOR RESPONSE

Reviewer 1: Dr Tim Taylor, University of Exeter Medical School, Truro, UK.

This is an interesting paper outlining a protocol for a study on the impact of a social housing intervention in Brazil. The study uses secondary data analysis of the novel merger of datasets from a large cohort study, a major social housing intervention and health datasets available in the country. The data analysis protocol appears robust, with appropriate plans for the identification of comparable individuals using appropriate data on location and socioeconomic status.

We thank you for your observation.

1. Page 3 – lines 10-15. I am always a bit nervous about “firsts”. Here you are claiming to be the “first study to evaluate the effect of a social housing programme on health outcomes at national level”. I don’t dispute that what you are planning is important and that it is novel, but I wonder about this statement. There is work using the SAIL database in Wales on certain interventions that have been made on housing – some of which largely fall in the area of improving social housing – e.g. the Housing Regeneration and Health Study. Admittedly this study focuses on improving existing social housing as the intervention, rather than the creation of new housing. There are also studies in Australia (e.g. Bentley et al, 2018) on mental health and social housing. A systematic review of the literature from 1887 to 2007 shows some studies that cover the provision of new housing (Thomson et al, 2009). While what you are doing is a major advance, I wonder if you might focus on scale and the low- to mid-income focus in terms of the novelty of the study on lines 10-16.

We agree that we should highlight the advance that our study proposes, with a focus on scale of the programme and the fact that our study will improve the discussion related to the effect of social housing programmes on health outcomes in low-and middle income countries, such as Brazil. We appreciate the suggested references, and we added those that were not yet included in our protocol. We have rephrased the relevant sentence (Please see section “Strengths and limitations of this study”, Line 80-83, page 3):

“This will be the first study to evaluate the effect of a major social housing programme on health outcomes in a middle-income country and is likely to be the largest of its type across the world. This allows us to assess impacts on population subgroups from an intersectionality perspective.”

2. You suggest there is a lack of patient and public involvement in the research (Page 5, lines 42-47). I am surprised you are not including any public involvement in the research process – for dissemination if not for research design. Do your stakeholders mentioned in line 39 of the Ethics summary not include the public or representative groups? If there is truly no public engagement then it might be worth putting something in lines 42-47 on why this is (perhaps funding constraints?) and it might be worth thinking of attempting to obtain further funding to support such engagement. It is all too easy to analyse data on households without engaging with the people in them, which can give a richer understanding of the reasons why the relationships are as they come out or of what questions could be posed in the research.

In contrast to the UK, patient and public involvement remain rare in Brazil and the governance procedures for including the public as a partner in research are not in place. Cidacs team made an effort to had several meetings with MCMV managers and consulting MCMV specialists who implemented the programme in Brazil, both at the national and subnational level. Unfortunately, the politically sensitive nature of the housing programmes means that policy stakeholders may be less willing to participate alongside public members.

We do not have permission to contact individual MCMV recipients, but we are working with democratically elected public representatives as part of our policy stakeholders. We compromise to disseminate the findings to the public. We hope in the future to undertake qualitative research which will provide a more nuanced understanding of the mechanisms through which any observed impacts operate and if we are able to contact individual recipients, we will also seek to involve them in the research itself too.

As you note, we have been engaging closely with policy stakeholders and the protocol has been devised as a joint effort with the National Housing Secretariat from the Ministry of Regional Development, in order to guarantee that the findings would answer relevant policy questions. CIDACS staff are, in synergy with these key stakeholders and decision-makers, providing the methodological rigor needed to assure sound results. These will, in turn, be able to be incorporated in the National Housing Plan, which is currently under development in Brazil. The National Housing Secretariat from the Ministry of Regional Development will not interfere in the analysis and results from studies planned in this protocol. We updated the information related to this topic in the protocol (Please see section “Methods and analysis/Ethics summary, Line 163-175 Page 5-6):

“Patient and public involvement:

This research was done without public involvement. Public were not invited to comment on the study design and were not consulted to develop public relevant outcomes or interpret the results, since we use an administrative and deidentified dataset and do not have permission to contact individuals. Study findings will be discussed with managers and specialists from the National Housing Secretariat from the Ministry of Regional Development and the published results will be disseminated to the public through the mass media. This study is a joint effort with the National Housing Secretariat from the Ministry of Regional Development, in order to guarantee that the findings would answer relevant policy questions. CIDACS staff are, in synergy with these key stakeholders and decision-makers, providing the methodological rigor needed to assure sound results. Findings will be incorporated into the National Housing Plan which is currently under development in Brazil. The National Housing Secretariat from the Ministry of Regional Development will not interfere in the analysis and results from studies planned in this protocol.”

3. P7 – I am wondering if you might consider increasing the consideration of potential short term negative impacts of the programme – e.g. if relocation leads to increased stress, fragmentation of social networks (which you do mention briefly) and the wider consequences for households of such a

move. I think you may also be being optimistic that reduced housing costs would lead to disposable income being spent on healthy food.

We acknowledge the possibility of short term negative impacts of the programme on health and wider outcomes. In order to support our initial hypothesis - receiving a new house will improve health and wellness of vulnerable families –, we focus our logic model on possible positive effects of new houses on health, but we mentioned possible negative outcomes related to the loss of social networks (pp6-7) Since the literature does not provide a consensus related to the short-term effect of relocation on health outcomes, we highlight both options in our logic model (Please see section “Methods and analysis/Logic models Line 229-231, page 7). We have now added an additional sentence to highlight these potential negative impacts within the text too:

“In addition, reducing the cost of housing also provides greater access to resources which can be spent on healthier food and health care, leading to better control of cardiometabolic risk factors [31,32]. In contrast, it is possible that greater availability of income could lead to greater consumption of unhealthy products (such as ultra-processed food, tobacco and alcohol) which could in turn increase cardiovascular risk (Martins & Monteiro, 2016; Sperandio et al 2017) .”

References:

Martins AP, Monteiro CA. Impact of the Bolsa Família program on food availability of low-income Brazilian families: a quasi experimental study. *BMC Public Health*. 2016 Dec 1;16(1):827.

Sperandio, N., Rodrigues, C. T., Franceschini, S. D. C. C., & Priore, S. E. (2017). The impact of the Bolsa Família Program on food consumption: a comparative study of the southeast and northeast regions of Brazil. *Ciência & Saúde Coletiva*, 22, 1771-1780.

Minor comments

1. 1. Page 3 Line 16 – meaning of “intersectionality” not clear to me. “Intersectoral” or “interdisciplinary” might be better words.

In line 16 we use "intersectionality" in the sense proposed by Crenshaw (1990), which has been used in studies on social epidemiology to capture how the inequalities of race, gender, and class among others, as well as their interaction (e.g. race x gender x class) influence health outcomes. In the Brazilian context, it is important to consider the intersectionality framework to improve understanding of the distribution of health outcomes, and to analyze health disparities among social groups. The large sample size of our data (114 million people or 55% of the Brazilian population) will allow us to apply this framework and to explore health disparities among social groups, considering race/ethnicity, gender and class, as well as considering the intersection between those multiple social markers of health inequalities (Please see section “Strengths and limitations” Line 84-85, page 3).

Reference: Crenshaw, Kimberlé. "Mapping the margins: Intersectionality, identity politics, and violence against women of color." *The legal response to violence against women* 5 (1997): 91.

2. I think you are overstating weakness on longer term outcomes for bullet 5 (Page 3, line 31) – to have up to 8 years is not exactly limited (often studies are shorter in their focus than this in terms of follow-up). I would consider rephrasing this as a strength.

We have rephrased this issue. We understand that 8 years is not exactly limited, but since we are working with chronic diseases, such as CVD mortality, and a neglected disease with a long term of incubation, such as leprosy, we choose to be more cautious in our initial approach,

and highlight that we might not be able to estimate the long term effect at the moment. Reviewer 2 notes that an 8 year follow up “is a short time window to examine the chronic pathway under consideration”. We have rephrased the bullet point in order to be clearer (Please see section Strengths and limitations of this study Line 93-95, page 3):

“Finally, this study does not allow estimation of long-term effects of MCMV on health, especially for NCDs, such as CVD mortality, and neglected diseases, such as leprosy, given the limited length of follow-up available (up to eight years).”

3. Page 4 Line 37 – add “and infectious disease, such as leprosy and tuberculosis”

We have rephrased the information in this sentence (Please see Line 127, page 4):

“Taking into account these relationships, there is a policy expectation that housing interventions could contribute to improve health and reduce social inequalities, especially among the most vulnerable [1,5,12]. Despite this, we are aware of little or no robust evidence on the positive and negative effects of housing conditions on cardiovascular disease (CVD) and infectious diseases, such as leprosy and tuberculosis. Understanding housing impacts on health in low and middle-income countries also remains particularly poorly understood.”

4. Page 5, line 24 insert “and” at end of line

We have corrected the sentence (Please see page 5).

Reviewer 2: Rebecca Bentley, University of Melbourne, Australia.

This study protocol for evaluating the effect of a social housing program in Brazil using a natural experiment methodology will enable quantification of the health protective potential of social housing. This is an important and worthwhile area of investigation that is difficult to research because of the composition of residents in social housing often being different to other tenures. It proposes to use cohort data, that includes information on tenancy, deterministically linked to registry data to examine time spent in social housing in relation to cardiovascular mortality, leprosy and tuberculosis (chosen to cover both chronic and infection disease pathways). The cohort in social housing will be matched to similar individuals in other housing tenures using propensity score approaches. The proposed project will make efficient use of existing cohort and registry data. Importantly, because of the large sample size, 114 million people or 55% of the Brazilian population, the experiences of subgroups defined to capture the complexity of identity (intersectionality) can be explored. There is a lack of information on important behavioural risk factors (e.g. smoking), however, these factors (while correlated with social housing) are unlikely to determine access to the program. Authors acknowledge that the maximum follow-up is 8 years, which is a short time window to examine the chronic pathway under consideration. Pathways are explicitly described in logic models, however, these models (unlike directed acyclic graphs) do not identify sources of confounding, the direction of relationships, mediators and modifiers which would be useful for framing the approach more clearly (particularly if complex sources of heterogeneity are to be explored). Sensible robustness checks and sensitivity analyses are proposed.

We appreciated your observation and comments.

We considered the DAG approach a valuable tool for identify confounders and other sources of bias, such as collider variables. However, DAGs are not always that helpful for demonstrating modifiers since scale is not explicitly shown in DAGs. Furthermore, DAGs should ideally be specific to a single exposure (in our case MCMV) and each outcome of interest. This would therefore mean that we

would ultimately require many more DAGs and given that we have pre-defined covariates available which are not time-varying, we feel that DAGs would actually be less useful than logic models to inform the analysis. By drawing on logic models, we are better able to illustrate mechanisms through which the intervention might operate and key moderators. Furthermore, the logic models have informed the development of our analysis plans and we would therefore prefer to retain them in preference of DAGs, since they more accurately reflect the theoretical underpinning of our analyses.